# Adipose-Derived Mesenchymal Stem Cells Protect Endothelial Cells from Hypoxic Injury by Suppressing Terminal UPR In Vivo and In Vitro

**DOI:** 10.3390/ijms242417197

**Published:** 2023-12-06

**Authors:** Michael Keese, Jiaxing Zheng, Kaixuan Yan, Karen Bieback, Benito A. Yard, Prama Pallavi, Christoph Reissfelder, Mark Andreas Kluth, Martin Sigl, Vugar Yugublu

**Affiliations:** 1Department of Surgery, Medical Centre Mannheim, Medical Faculty Manheim, Heidelberg University, 68167 Mannheim, Germany; m.keese@theresienkrankenhaus.de (M.K.); jiaxing.zheng@medma.uni-heidelberg.de (J.Z.); kaixuan.yan@medma.uni-heidelberg.de (K.Y.); prama.pallavi@medma.uni-heidelberg.de (P.P.); christoph.reissfelder@umm.de (C.R.); 2European Center of Angioscience (ECAS), Medical Faculty Manheim, Heidelberg University, 68167 Mannheim, Germany; 3Department for Vascular Surgery, Theresienkrankenhaus Mannheim, 68165 Mannheim, Germany; 4Institute of Transfusion Medicine and Immunology, Medical Faculty Manheim, Heidelberg University, 68167 Mannheim, Germany; karen.bieback@medma.uni-heidelberg.de; 5V Department of Medicine, Medical Faculty Manheim, Heidelberg University, 68167 Mannheim, Germany; benito.yard@medma.uni-heidelberg.de; 6DKFZ-Hector Cancer Institute, Medical Faculty Mannheim, Heidelberg University, 68167 Mannheim, Germany; 7RHEACELL GmbH & Co. KG, Im Neuenheimer Feld 517, 69120 Heidelberg, Germany; andreas.kluth@rheacell.com; 8Department of Cardiology, Angiology, Haemostaseology and Medical Intensive Care, University Medical Centre Mannheim, Medical Faculty Mannheim, Heidelberg University, 68167 Mannheim, Germany; martin.sigl@umm.de

**Keywords:** peripheral artery disease, adipose-derived stem cells, hypoxia, critical limb ischemia, unfolded protein response, endoplasmic reticulum

## Abstract

Adipose-derived stem cells (ASCs) have been used as a therapeutic intervention for peripheral artery disease (PAD) in clinical trials. To further explore the therapeutic mechanism of these mesenchymal multipotent stromal/stem cells in PAD, this study was designed to test the effect of xenogeneic ASCs extracted from human adipose tissue on hypoxic endothelial cells (ECs) and terminal unfolded protein response (UPR) in vitro and in an atherosclerosis-prone apolipoprotein E-deficient mice (ApoE^−/−^ mice) hindlimb ischemia model in vivo. ASCs were added to Cobalt (II) chloride-treated ECs; then, metabolic activity, cell migration, and tube formation were evaluated. Fluorescence-based sensors were used to assess dynamic changes in Ca^2+^ levels in the cytosolic- and endoplasmic reticulum (ER) as well as changes in reactive oxygen species. Western blotting was used to observe the UPR pathway. To simulate an acute-on-chronic model of PAD, ApoE^−/−^ mice were subjected to a double ligation of the femoral artery (DLFA). An assessment of functional recovery after DFLA was conducted, as well as histology of gastrocnemius. Hypoxia caused ER stress in ECs, but ASCs reduced it, thereby promoting cell survival. Treatment with ASCs ameliorated the effects of ischemia on muscle tissue in the ApoE^−/−^ mice hindlimb ischemia model. Animals showed less muscle necrosis, less inflammation, and lower levels of muscle enzymes after ASC injection. In vitro and in vivo results revealed that all ER stress sensors (BIP, ATF6, CHOP, and XBP1) were activated. We also observed that the expression of these proteins was reduced in the ASCs treatment group. ASCs effectively alleviated endothelial dysfunction under hypoxic conditions by strengthening ATF6 and initiating a transcriptional program to restore ER homeostasis. In general, our data suggest that ASCs may be a meaningful treatment option for patients with PAD who do not have traditional revascularization options.

## 1. Introduction

Critical limb ischemia (CLI), the most severe manifestation of peripheral arterial disease (PAD), is characterized by rest pain, ischemic ulceration, non-healing wounds, and foot gangrene, and is associated with a 6-month mortality rate of 20% and a 5-year mortality rate of 50% [1,2]. In addition, CLI is associated with a high risk of lower limb amputation, in approximately 10% to 40% of patients [3]. The treatment of CLI usually involves a combination of medical management, minimally invasive procedures, and surgical interventions to improve blood flow to the affected limb, relieve pain, promote wound healing, and prevent amputation [4,5]. Endothelial cell (EC) dysfunction is one of the key mechanisms contributing to tissue ischemia in CLI [6]. As a potential treatment for CLI that addresses EC dysfunction, ASCs have shown promise, but further research is required to gain a deeper understanding of the mechanisms involved and to determine the best administration, dosing, and outcomes for CLI [7].

Both in vivo and in vitro studies have shown that ASCs can improve healing rates and shorten healing times in critical limb ischemia [8,9,10]. ASCs are mesenchymal stromal cells (MSCs) that are obtained from abundant adipose tissue, adherent on plastic culture flasks, can be expanded in vitro, and have the capacity to differentiate into multiple cell types [11]. Although discovered only 20 years ago, stem cell therapy using ASCs has been extensively used to treat several diseases [12]. ASCs have recently gained significant attention in the realm of regenerative medicine due to their accessibility, abundance, and multilineage differentiation potential [13]. Clinically, ASCs have been explored for various applications. One of the hallmark applications of ASCs is in wound healing, where their ability to modulate inflammation and enhance angiogenesis has been harnessed to promote tissue regeneration [14]. In orthopedics, ASCs have shown promise in the treatment of osteoarthritis, demonstrating potential for cartilage repair and anti-inflammatory effects [15]. Cardiovascular medicine has also tapped into the potential of ASCs for treating conditions like myocardial infarction, given their capacity to promote neovascularization and inhibit cardiomyocyte apoptosis [16]. Furthermore, ASCs are being investigated for their immunomodulatory properties in graft-versus-host disease and autoimmune disorders [17].

In a hypoxic environment, EC dysfunction and endoplasmic reticulum (ER) stress are closely related and can contribute to the progression of the disease [18]. A better understanding of the link between ER stress and endothelial dysfunction in CLI opens up potential therapeutic avenues, which could help reduce endothelial dysfunction and thus help to restore blood flow [19]. Hypoxia disrupts calcium homeostasis and induces ER dysfunction and protein folding [20]. Consequently, the accumulation of numerous misfolded proteins in the ER lumen generates ER stress [18]. Eukaryotic cells have evolved an unfolded protein response (UPR) to ensure the fidelity of protein folding and prevent the accumulation of unfolded or misfolded proteins [21]. Mammalian cells recruit three UPR signaling cascades, initiated by three ER-localized protein sensors: inositol-requiring enzyme 1α (IRE1α), double-stranded RNA-dependent protein kinase (PKR)-like ER kinase (PERK), and activating transcription factor 6 (ATF6) [19,22,23]. Under normal physiological conditions, these sensors are associated with abundant ER chaperone BIP (immunoglobulin-heavy-chain-binding protein, also known as HSPA5 and GRP78) and remain inactive [19]. Their activation reduces the accumulation of unfolded proteins and accelerates the rate of protein folding in the ER lumen [22]. The downstream transcriptional programs of the UPR act to restore proteostasis. IRE1α increases the protein-folding capacity of the ER, assists in endoplasmic-reticulum-associated protein degradation (ERAD), and limits the entry of new protein molecules (regulated Ire1-dependent decay [RIDD]); PERK induces the transcription of genes and escapes translation inhibition under ER stress; and ATF6 increases the ER capacity to activate transcriptional programs and direct the misfolded proteins for degradation (ERAD) [19,22,23]. UPR-induced genes include those that increase the protein-folding capacity of the ER and mediate its expansion by increasing the biogenesis of ER and lipid components [23]. However, if restoration of proteostasis fails and ER stress is unabated, the UPR signaling switches to a pro-apoptotic mode, a process known as the terminal UPR [22]. In light of all these data, we investigated if and how ASCs modulate hypoxia-induced ER stress by regulating Ca^2+^ homeostasis in vitro and in vivo.

In view of this, our current study first uses both an in vitro assay and a murine in vivo model of acute-on-chronic ischemia to investigate the role of ASCs on potential hypoxic injuries to endothelial cells. Moreover, we especially focus on the role of UPR and Ca^2+^ homeostasis in endothelial cells.

## 2. Results

### 2.1. ASCs Enhance Angiogenic Properties of Hypoxic HUVECs In Vitro

Using an in vitro hypoxia model, we evaluated the effect of ASCs on the angiogenic properties of HUVECs assessing migration and tube formation, with three replicates for each treatment. Treatment with CoCl_2_ reduced the migration rate of HUVEC as compared to non-treated controls (P_CoCl2 vs. control_ = 0.0056) (Figure 1A). Culture in ASC-conditioned medium (ASC-CM) and co-culture with ASCs (ASC-CO) both restored the migratory capabilities of hypoxic HUVECs; however, the latter was more effective (P_ASC-CM vs. CoCl2_ = 0.0411; P_ASC-CO vs. CoCl2_ = 0.0027, P_ASC_-_CO vs. ASC-CM_ = 0.0279) (Figure 1A). Likewise, in the tube formation assay, CoCl_2_ treatment significantly reduced the number of grids and connection points (P_CoCl2 vs. control_ = 0.0249) (Figure 1B). Furthermore, ASC-CM and ASC-CO significantly increased the number of meshes and junctions under CoCl2 treatment; however, the former was more effective (P_ASC-CM vs. CoCl2_ =< 0.001; P_ASC-CO vs. CoCl2_ = 0.3125; P_ASC-CO vs. ASC-CM_ = 0.2609) (Figure 1B).

In addition to the migratory ability of HUVECs, the balance between ROS generation and clearance and cell viability are crucial factors in angiogenesis. MTT analysis revealed that CoCl_2_ significantly reduced the cell viability of HUVECs (P_CoCl2 vs. control_ < 0.001) while ASC-CM and ASC-CO reversed this effect (P_ASC-CM vs. CoCl2_ < 0.001; P_ASC-CO vs. CoCl2_ < 0.0001; P_ASC-CO vs. ASC-CM_ = 0.4586) (Figure 1C). In parallel, there was a significant increase in ROS levels in rogfp-3 expressing HUVECs following treatment with CoCl_2_ (P_CoCl2 vs. control_ < 0.0001). However, culture with ASCs and culture in ASCs- CM reversed this effect (P_ASC-CM vs. Control_ = 0.0002; P_ASC-CO vs. Control_ = 0.0006). ROS levels returned to the baseline faster with ASC-CM (Figure 1D).

### 2.2. ASCs Treatment Protect Hindlimb Muscles against Hypoxia-Induced Injury

A Western diet led to plaque formation in both control and treatment groups (Figure 2A). No differences in plaque formation could be observed after scarification (Appendix A). Double ligation of the femoral artery was successfully performed on all mice both in the control (N = 12) and treatment group (N = 12). Two mice died in the control group before seven days, while no mice died in the ASCs treatment group (P_Control vs. ASC-treated_ = 0.1592) (Figure 2B). Furthermore, after the ASCs treatment, the mice showed significantly less weight loss compared to that shown by the control (P_Control vs. ASC-treated_ = 0.0381) (Figure 2C).

Treatment employing xenogeneic ASCs extracted from human adipose tissue protected the hindlimb muscle tissue; this is shown by significantly less Lactate dehydrogenase (LDH) and significantly higher amounts of Mb found in the VL tissue of ASC-treated mice compared to controls (LDH, P_Control_ vs. ASCs_treated_ = 0.0097; Myoglobin, P_Control_ vs. _ASC- treated_ = 0.0438) (Figure 3C,D). Furthermore, ischemic hindlimbs exhibited significantly more CD31-positive areas, a surrogate for microvascular density, than did the control mice following treatment with ASCs (P_Control_ vs. ASCs_treated_ < 0.0001) (Figure 3B). Consequently, our in vivo experiments showed that human ASCs improved muscle resistance to ischemia-induced damage, illustrating the potential of xenogeneic cell therapy in regenerative medicine as well.

### 2.3. ASCs Restore ER Calcium and Thereby Prevent ER Stress

Hypoxia-led oxidative stress causes deregulation in Ca^2+^ homeostasis. Therefore, we evaluated the effect of CoCl_2_ treatment on cytoplasmic and ER Ca^2+^ in HUVECs. The Fura-2 ratio of the HUVECs increased significantly after 4 h of treatment with 80 µM CoCl_2_ compared to that of controls (P_CoCl2 vs. control 4 h_ < 0.0001). Incubation with ACSs-CM brought the cytoplasmic Ca^2+^ in hypoxic HUVECs to that of non-treated control over a period of 12 h (P_ASC-CM vs. control 12 h_ > 0.6059). Under co-culture with ASCs restoration of the cytoplasmatic Ca^2+^ took nearly 14 h longer (P_ASC-CM vs. control 28 h_ = 0.9736) (Figure 4A). Changes in cytoplasmic Ca^2+^ impair ER calcium homeostasis; therefore, we also assessed this in our in vitro model. The FRET ratio (YFP/CFP) decreased significantly upon CoCl_2_ treatment in D1ER expressing HUVECs (P_CoCl2 vs. control 4 h_ < 0.0001) indicating depletion of ER Ca^2+^. Both ASC-CM and ASC-CO treatments restored ER Ca^2+^ as shown by an increase in FRET ratio (P_ASC-CM vs. control 4 h_ < 0.0001). Here as well, ASC-CM (P_ASC-CM vs. control 8 h_ = 0.0002, P_ASC-CO vs. control 8 h_ < 0.0001) was faster than ASC-CO (P_ASC-Co vs. control 20 h_ = 0.0059) (Figure 4C). We also performed acceptor photo bleaching using fixed cell samples, such as the live cell experiments; correspondingly, we observed a decrease in the FRET ratio upon treatment with CoCl_2_ (P_CoCl2 vs. control_ < 0.0001). In HUVECs treated with ASC-CM and ASC-CO, the FRET ratio normalized back to the baseline levels (P_ASC-CM vs. control 8 h_ = 0.7406, P_ASC-CO vs. control 8 h_ = 0.0123) (Figure 4B).

Since the depletion of ER Ca^2+^ stores resulted in ER stress and activated the UPR, we further checked the expression of UPR proteins in vitro. CoCl_2_ treatment caused high expression of BIP (P_CoCl2 vs. control_ < 0.0001) (Figure 5B), ATF6 (P_CoCl2 vs. control_ = 0.0004) (Figure 5C), and XBP1 (P_CoCl2 vs. control_ = 0.0004) (Figure 5D). No significant difference was observed in the expression of CHOP: CoCl_2_ did not lead to a higher expression. Expression also remained largely unchanged under the treatment of tunicamycin (Figure 5E). The co-culture with ASCs and the ASC-conditioned medium led to a reduced expression of these factors: BIP (P_ASC-CM vs. control_ = 0.1429, P_ASC-CO vs. control_ = 0.0673) (Figure 5B), ATF6 (P_ASC-CM vs. control_ = 0.3885, P_ASC-CO vs. control_ = 0.2214) (Figure 5C), and XBP1(P_ASC-CM vs. control_ = 0.4610, P_ASC-CO vs. control_ = 0.4877) (Figure 5D). Although there was an increase in the expression of protein from the UPR pathway, the concentration of CoCl_2_ we used did not lead to cell apoptosis.

Afterward, to correlate these findings with the in vivo data, we determined the expression levels of BIP, ATF6, XBP1, and CHOP in the total protein isolated from the muscle specimens collected from both hind limbs of the treated and control mice. The expression levels of BIP (P_control vs. ASC_ = 0.0005) (Figure 6B), ATF6 (P_control vs. ASC_ = 0.0044) (Figure 6C), XBP1 (P_control vs. ASC_ = 0.0358) (Figure 6D), and CHOP (P_control vs. ASC_ = 0.0015) (Figure 6E), were lower in the treatment group than in the control group.

## 3. Discussion

In this study, we have demonstrated that in the ApoE^−/−^ PAD mice model, the xenogeneic ASC treatment improved functional recovery within seven days after surgery. A week after the generation of critical limb ischemia, the ASC-treated mice lost significantly less weight in comparison to the control mice. The histological analysis of the microvascular density in bilateral gastrocnemius muscle of ASC-treated mice showed enhanced angiogenesis. Considering that capillaries supply the nutrients (oxygen, glucose, etc.) and dispose of cellular waste products, microvascular density (MVD) is a crucial prognostic factor for PAD [24]. Patients with PAD and poor prognosis usually show low MVD. In our ApoE^−/−^ PAD mice model, the MVD ratio of the ASC treatment group was significantly higher than that of the control group, indicating that ASC treatment enhanced angiogenesis. ASC therapy–induced neovascularization is conducive to restoring metabolic homeostasis, thereby improving functional recovery. Additionally, the average LDH levels of the muscles in the ASC group were lower than those of the muscles in the control group; moreover, these results were statistically significant, indicating that ASCs could improve metabolic homeostasis. In this study, we measured LDH in homogenized muscle tissues rather than the conventional serum-based approach. This method allowed us to directly assess muscle cell damage at the tissue level, capturing immediate LDH releases from injured cells. Moreover, the use of xenogeneic transplantation in our study, injecting human ASCs into mice, bridges the critical gap in interspecies cell therapy dynamics and its therapeutic implications, enabling novel approaches in regenerative medicine.

To study the underlying cell biological mechanisms, we established an in vitro hypoxic environment that reflected the levels of cellular, biochemical, and molecular hypoxia responses. We applied a chemically induced hypoxia model instead of a gas model which requires a hypoxic chamber being aware that this can mimic only a minor part of gas hypoxia [25]. A commonly used chemical to induce hypoxia is CoCl_2_, which strongly stabilizes HIF-1α and HIF-2α under normoxic conditions for several hours in a dose- and time-dependent manner [26].

In the mimicked hypoxic environment, the MTT analysis revealed that CoCl_2_ significantly reduced the cell viability of HUVECs, whereas ASC-CM and ASC-CO reversed this effect, indicating that ASCs promoted the metabolic activity of ECs under hypoxia. Migration assays revealed that CoCl_2_ reduced the migration rate. Again, ASCs indirectly or directly accelerated the migration rate of ECs in the ASC-CO group. The results of the endothelial tube formation assay showed that the number of meshes, and junctions was significantly reduced following CoCl_2_ treatment. Correspondingly, the effect was significantly reversed by ASC-CM and ASC-CO following CoCl_2_ treatment. Furthermore, CoCl_2_ increased the levels of ROS; this effect was reversed by ASC-CM and ASC-CO therapy in 24 h. The therapeutic effect of ASC-CM was exerted earlier than that of ASC-CO. ASC-CO conditioned medium produced delayed effects, which may be attributed to the time taken by the ASCs to exert a paracrine effect on ECs in the co-culture.

Thus, in summary, ASCs accelerated the migration, tube formation ability, and metabolic activity of ECs while they reduced the production of ROS under hypoxia. This indicates that they may promote the angiogenesis ability of ECs. Our results therefore showed that ASCs, directly or indirectly, promoted the survival and the angiogenic functions of ECs under hypoxia.

ER, a ubiquitous organelle, is responsible for the synthesis, proper folding, maturation, and assembly of proteins before these are further processed by the Golgi apparatus. Stable ER Ca^2+^ concentration or homeostasis is crucial to maintaining cellular functions. ER Ca^2+^ depletion causes misfolding or unfolding of proteins, resulting in their accumulation within the ER lumen. This in turn causes ER stress and activates the UPR [27,28]. UPR is a normal adaptive and protective mechanism to reduce the rate of protein synthesis, increase the folding ability of proteins, and help misfolded or unfolded proteins to enter cellular degradation pathways [23]. Nevertheless, non-resolved ER stress can cumulatively cause cell death [19,22]. Mungai et al. and Gusarova et al. demonstrated the existence of this pathway under hypoxia in osteosarcoma cells and alveolar epithelial cells [29,30]. We therefore investigated how CoCl_2_-induced hypoxia affects the Ca^2+^ homeostasis in HUVECs and how this is thereafter regulated by ASCs. It is well known that hypoxia reduces the ER Ca^2+^ restoring ability by increasing the release of ER Ca^2+^. In parallel, an influx of extracellular Ca^2+^ is induced, consequently increasing the cytoplasmic Ca^2+^ concentrations [31]. The overloading of cytoplasmic Ca^2+^ subsequently induces cell dysfunction and apoptosis, indicating prolonged or severe hypoxia [21]. The imbalance in the interaction between cytoplasmic Ca^2+^ and ER Ca^2+^ is a sign of ER stress, which may affect the survival of the endothelium.

We used fluorescent sensor assays and functional microscopy to analyze the FRET ratio to detect Ca^2+^ levels. After 4 h of CoCl_2_ treatment, the HUVECs had a significantly reduced FRET ratio, reflecting a decrease in ER Ca^2+^ under hypoxia. ASC-CM and ASC-CO treatment for 24 h reversed this effect, with the therapeutic effect of ASC-CM occurring earlier than that of ASC-CO. The results of FRET microscopy showed that the FRET ratio of HUVECs treated with CoCl_2_ was significantly lower than that of control cells. Dynamic changes in cytoplasmic Ca^2+^ were detected by fluorescence using Fura-2AM. Compared with the control group, the Fura-2 ratio of the CoCl_2_ group increased significantly after 4 h of CoCl_2_ treatment due to CoCl_2_-induced elevation in cytoplasmic Ca^2+^. However, the therapeutic effect of ASCs reduced the ratio in the following 24 h, although the impact from ASC-CM was displayed earlier than that of ASC-CO. These results implied that CoCl_2_ mimicking a hypoxic pathological condition of ECs results in a continuous decrease in ER Ca^2+^ and an increase in cytoplasmic Ca^2+^ due to reduced ER Ca^2+^ restoring ability. The ASC-conditioned medium and adipose co-culture group reversed the above calcium imbalance under the same conditions. These results also indicated that stem cells regulate ER stress.

Hypoxic stress induces global gene expression changes by altering the cell’s metabolic and angiogenic pathways and restoring oxygen homeostasis, thereby promoting cell survival [32,33]. A failure of these repair and adaptive mechanisms causes cells to modify their gene expression profiles and induce programmed cell death [32,34,35]. However, these changes are accompanied by the deregulation of mitochondrial and ER functions, reflected by perturbations in protein folding and trafficking. Erratic protein folding activates another specific stress response pathway; the UPR promotes cellular survival by restoring endoplasmic and mitochondrial homeostasis via distinct signaling networks [21]. Critical changes in mitochondrial functions occur during hypoxia, causing elevated ROS levels. Furthermore, the proper folding of mitochondria-encoded, as well as the import and corresponding refolding of mitochondrial nucleus-encoded proteins, is crucial for the proper functioning of this organelle. Hence, prolonged hypoxia eventually results in perturbations in mitochondrial protein folding and activation of a related specific stress response mechanism known as the mitochondrial UPR. To better understand the role of UPR in HUVECs under hypoxia we here investigated the expression of BiP, which initiates the UPR by dissociating three proteins, namely PERK, IRE1α, and ATF6, from the ER lumen. Our in vitro and in vivo results revealed that all ER stress-associated proteins (BIP, ATF6, CHOP, and XBP1) are activated once tunicamycin is processed. However, only BIP, ATF6, and XBP1 are activated under hypoxia. We observed that the expression of these proteins was reduced in the stem cell treatment group. Stem cells effectively alleviated endothelial dysfunction under hypoxic conditions by strengthening ATF6 and initiating a transcriptional program to restore ER homeostasis, induce BIP expression, promote protein chaperones and lipid synthesis, stimulate ER degradation, and enhance N-glycosylation of XBP1 by increasing the ER’s folding capacity, as well as increasing the expression of chaperones and proteins involved in ER-associated degradation (by ER degradation-enhancing α-mannosidase-like protein [EDEM]) and vesicular trafficking. The inactivation of CHOP could potentially be attributed to the fact that the CoCl_2_ concentration used was insufficient to activate it, leading to cell death.

In summary, we first explored how ASC stems can mediate protection against hypoxic injury, specifically by investigating their role in suppressing terminal UPR. Our findings with human umbilical-vein endothelial cells (HUVECs) are in line with previous studies that demonstrate the therapeutic potential of stem cells in ameliorating hypoxic conditions [36]. The utilization of 80 μM Cobalt (II) chloride (CoCl_2_) in our experiments to induce a hypoxic environment, and the subsequent protective effects observed in stem cells, are consistent with the findings of Ejtehadifar et al. (2015) [37]. Their study explored the influence of hypoxic conditions on mesenchymal stem cell biology, highlighting the adaptive and protective responses of these cells in low-oxygen environments. This similarity in findings underscores the robustness of mesenchymal stem cells’ protective mechanisms under hypoxia-induced stress conditions [37]. Furthermore, the interplay between ER Ca^2+^ dynamics and angiogenesis has previously been hinted at, but our findings offer a more direct link [38]. In future works, it will be also intriguing to apply an ASC-conditioned medium into ischemic tissues using a similar in vivo setup.

Overall, our study does offer a fresh potential perspective on the adaptability and reliability of adipose-derived mesenchymal stem cells in mitigating cardiovascular maladies.

### Limitations

First, we did not use hypoxic chambers to simulate true hypoxia; instead, we used CoCl_2_ to mimic hypoxic conditions. Therefore, our results are potentially incomparable with those observed in a truly hypoxic environment. An in vitro hypoxic model could be established using a hypoxic chamber in the future to overcome this shortcoming. Second, we have so far not correlated our results with potential paracrine effects of stem cells, such as growth factors and VEGF tracking, on ECs (which will be the topic of ongoing studies). Third, to investigate the relationship between ER Ca^2+^ recovery ability and EC angiogenesis function using D1ER and Fura-2 AM, we detected the dynamic changes in ER Ca^2+^ and cytoplasmic Ca^2+^ in only in vitro and due to the nature of these sensors—not in vivo. Thus, more studies are warranted in the future to further explore and explain the therapeutic mechanism of stem cells in more depth. Finally, it will be also intriguing to investigate the influence of an ASC-conditioned medium on the fate of ischemic muscular tissue.

## 4. Materials and Methods

### 4.1. Chemicals

All chemical reagents were purchased from Sigma Aldrich (Sigma-Aldrich Chemie GmbH, Munich, Germany) unless otherwise indicated. Treatment with 80 µM Cobalt (II) chloride (CoCl_2_) for a period of 4 h was used to mimic hypoxia in human umbilical-vein endothelial cells (HUVECs). Tunicamycin (TUN) 2.0 μg/mL was used to induce UPR in HUVECs.

### 4.2. Cell Culture

Human umbilical-vein endothelial cells (HUVECs) were isolated from freshly available umbilical cords provided by the Obstetrics Department of the University Hospital Mannheim. HUVECs isolation was approved by the local ethics committee (Medizinische Ethik-Kommission II, Medizinische Fakultät Mannheim, 2015-581N-MA, Mannheim, Germany) and was performed as previously described [39]. The cells were grown in basal endothelial cell growth medium (Provitro GmbH, Berlin, Germany) supplemented with 2% fetal bovine serum (Gibco™ Thermofisher Scientific, Sao Paulo, Brazil) without antibiotics. HUVECs from three different donors were pooled, and the experiments were conducted till passage 7 at approximately 80−90% visual confluence. We have mixed cells from every three donors into one group.

ASCs were isolated from the visceral fat as described by S. Kern et al. [40]. The adipose stem cells were extracted from human adipose tissue, and for each biological replicate samples from at least three independent donors were propagated for experimental use. The study was approved by the Mannheim Ethics Commission II (vote numbers 2010-262 N-MA, 2009-210 N-MA, 49/05 and 48/05). The cells were cultured in Dulbecco’s Modified Eagle’s Medium normal glucose (NG; 1 g/L, PAN Biotech; Aidenbach, Germany) supplemented with 10% FCS (Gibco™ Thermofisher Scientific, Brazil), 1% penicillin/streptomycin (Gibco™ Thermofisher Scientific, Frankfurt, Germany), and 2% L-glutamine (Gibco™ Thermofisher Scientific, Germany). Prior to use, all ASCs were assured to express the typical MSC immunophenotype and to differentiate into adipogenic and osteogenic lineages. The cells were then incubated with fluorescent-conjugated antibodies against CD73, CD90, CD105, and negative markers such as CD34 and CD45 for 30 min in the dark at 4 °C. Following the incubation, cells were washed and resuspended in PBS, and then subjected to flow cytometry analysis. Data acquisition was performed on flow cytometry (BD FACSCalibur) and data analysis was carried out using (FlowJo). ASCs were characterized by the positive expression of CD73, CD90, and CD105, and the negative expression of CD34 and CD45. Both HUVECs and ASCs were maintained at 37 °C in a 5% CO_2_-humidified atmosphere. We have mixed cells from every three donors into one group.

### 4.3. Co-Culture and Conditioned Medium Preparation

Co-culture experiments were performed using ThinCert™ cell culture inserts suitable for 6- and 24-well plates (Greiner Bio-One, Frankfurt, Germany). 2 × 104 HUVECs per cm^2^ were seeded in 6- and 24-well plates coated with 1% gelatin, and 0.6 × 104 ASC per cm^2^ were seeded into ThinCert™ cell culture inserts (4.5 cm^2^/insert, Greiner Bio-One, Germany, 657640) and 24-well plates (0.3 cm^2^/insert, Greiner Bio-One, Germany, 657640). Co-cultures with ASCs (ASC-CO) were performed with a medium consisting of an ASC culture medium and an HUVEC culture medium mixed at a 1:1 ratio. The co-cultures were maintained at 37 °C in a 5% CO_2_-humidified atmosphere [41,42,43,44]. For the ASC-conditioned medium (ASC-CM), ASCs were cultured until 70% confluency, the medium was discarded, and the cells were washed twice with PBS, followed by the addition of endothelial cell growth medium (2011101, Provitro, Germany) containing 1% penicillin and streptomycin. After 24 h, culture media were collected and centrifuged at 2500 rpm for 10 min at 4 °C and frozen at −80 °C for later use [45]. The experimental outline is described in Appendix A.

### 4.4. MTT Assay

2 × 104 HUVECs per cm^2^ in the log phase were collected and seeded on gelatin-coated 24-well plates. One-day post-seeding cells were first treated with CoCl_2_ for 4 h and afterward, either cultured in ASC-CM or co-cultured in ASC-CO for another 24 h. MTT dye solution (1 mg/mL in HUVECs medium) was added to the cells for 4 h, and formazan salt crystals were dissolved in 100 μL of MTT solvent (40% of Dimethyl sulfoxide (DMSO, HN47.1, Roth, Germany), 40% of 10%-sodium dodecyl sulfate (SDS, 0183.3, Roth, Germany), 20% of DPBS, and 1.2% of Acetic acid (6755.1, Roth, Germany)). The absorbance was read the next day at 540 nm (reference length = 630 nm) using a multi-mode microplate reader (Tecan Spark, Zurich, Switzerland).

### 4.5. Scratch Assay

2.6 × 104 HUVECs per cm^2^ were seeded into 12-well plates. After overnight incubation, the cells were pre-treated with 80 µM CoCl_2_ for 4 h. Using a 200 μL pipette tip, a single straight line was scraped. The wells were washed to remove lose and floating cells, and afterward, ASC-CM and ASC-CO were introduced with the continued presence of 80 µM CoCl_2_. Images were acquired using a camera attached to an inverted microscope (DM IRB; Leica, Berlin, Germany) at 0 h, 3 h, 6 h, 12 h, and 24 h. The gap was quantitatively evaluated using the AutoCellSeg as described previously [46].

### 4.6. Tube Formation

After 4 h of treatment with 80 µM CoCl_2_, the HUVECs were diluted with 1% FBS medium to 2.5 × 104 cells/mL and seeded into pre-coated 24-well plates (10,000 cells/well) in a Matrigel basement membrane matrix without phenol red (356237, Corning, NY, USA). Thereafter, the HUVECs were then, respectively, treated with ASC-CM or ASC-CO. After 6 h of incubation, images were acquired using an inverted microscope and analyzed using the Angiogenesis Analyzer software plugin (ImageJ version 1.5).

### 4.7. Virus Production and Transduction

#### 4.7.1. roGFP3

Lentivirus particles were produced as described previously [47]. HUVECs were then ftransduced with different concentrations of lentiviral particles for 48 h; the transduction efficiency was assessed by visualization of GFP. In transduced HUVECs, an optimal expression of GFP was observed at 1:100 dilution and thus this lentiviral concentration was used in further experiments. The stability of the expression over passages was also verified by qualitative assessment of GFP under the fluorescence microscope. In up to three passages after transduction, no evident GFP expression changes were observed.

#### 4.7.2. D1ER

D1ER, a genetically encoded FRET-based ER Ca^2+^ biosensor that contains enhanced yellow fluorescent protein (YFP: Ex 514 nm/Em 527 nm) and enhanced cyan fluorescent protein (CFP: Ex 430 nm/Em 474 nm), was used to study changes in ER Ca^2+^. It was synthesized by Genewiz from Merck Sigma Aldrich in pUC57 with BamHI and XbaI restriction sites. The pUC57-D1ER was further cloned into pHR’SIN-cPPT-SEW 15 via BamHI and XbaI restriction sites. Lentivirus particles were produced as previously described, and HUVECs were transduced. The ratio of fluorescence resonance energy transfer was calculated by dividing CFP intensity by YFP intensity both detected from D1ER (FRET ratio = YFP/CFP). This ratio represented the dynamic changes in ER Ca^2+^ concentration.

### 4.8. Protein Isolation and Western Blotting

Western blotting (WB) was performed to detect the protein expression. The total protein lysate from HUVECs and mice hindlimb muscle tissue was mixed with loading buffer (161-0767, Bio-Rad, Frankfurt, Germany), denatured by incubation at 100 °C for 10 min, loaded onto 10% SDS-PAGE gels, and electrophoresed at 200 V for about 35 min. Afterward, proteins were transferred onto polyvinylidene fluoride (PVDF, 1620177, Bio-Rad, Germany) membranes. Hereafter, the PVDF membranes were blocked for 1 h in Tris-buffered saline with 0.1% Tween 20 (TBST) containing 5% non-fat milk. Membranes were next incubated overnight at 4 °C with primary antibodies against GRP78 (BIP) (1:5000, PA5-34941, Invitrogen, Bartlett, IL, USA), XBP1 (1:1000, PA5-27650, Invitrogen, USA), ATF6 (1:1000, PA5-20215, Invitrogen, Bartlett, USA), GADD153 (CHOP) (1:1000, NB600-1335, Novus Biologicals), phospho-IRE1 alpha (1:1000, PA1-16927, Invitrogen, Bartlett, USA), IRE1 alpha (1:1000, PA1-16928, Invitrogen, Bartlett, USA), and β-actin (1:1000, ab8227, Abcam, Frankfurt, Germany). Following the incubation with primary antibodies, PVDF was washed thrice with TBST (10 min each time) and incubated for 1 h at room temperature with a corresponding horseradish peroxidase (HRP)-conjugated anti-rabbit (Sc-2357, 1:1000, Santa Cruz, München, Germany) or anti-mouse (Sc-51610, 1:1000, Santa Cruz) secondary antibody. The membrane was washed again thrice with TBST (5 min each time) following the detection of immune reactive protein bands using an enhanced luminol reagent (1 min–5 min exposure, Western Lightning Plus-ECL, PerkinElmer, Waltham, MA, USA, 203-17431) and visualized by chemiluminescence (1 min to 5-min exposure). To detect the phosphorylation of IRE1 alpha, the PVDF membrane was stripped using the following protocol: the stripping buffer with 2-mercaptoethanol (M6250, Sigma-Aldrich, Burlington, VT, USA) was heated to 65 °C, and the PVDF membrane was immersed in the stripping buffer for 35 min with agitation. Following this, the PVDF membrane was washed extensively for 10 min in TBST twice. Afterward, it was blocked for 1 h in TBST containing 10% non-fat milk. After stripping, the PVDF membrane was incubated with primary and secondary antibodies and subsequently exposed as described earlier. Densitometric analysis was performed using the NIH ImageJ software (v1.52, Bethesda, MD, USA). All the lanes were normalized to β-Actin; in addition to normalized to β-Actin, the right leg/left leg ratio was used for the WB of the in vivo part.

### 4.9. Measurement of ER Ca^2+^

HUVECs transduced with the D1ER sensor were seeded into 1% gelatin-coated 24-well-plates (3.8 × 104 cells per well). 24 h post-seeding cells were treated with 80 μM CoCl_2_ for 4 h. Thereafter, the cells were either co-cultured with ASCs or cultured in ASC-CM for 24 h. The fluorescence intensity of CFP and YFP was detected by a Tecan multimode microplate reader every 2 h. In addition to live cell measurements, ER Ca^2+^ was also measured by confocal microscopy. To this end, HUVECs transduced with a D1ER sensor were seeded into 1% gelatin-coated 15 mm coverslips in 24-well-plates (3.8 × 104 cells per well), and the experiment was set up as described above. After 24 h treatment, the cells were fixed with 4% paraformaldehyde (PFA) and imaged by an SP5 microscope system (Leica, Germany) to detect the fluorescence emitted by YFP and CFP. To compare the FRET ratio among the different groups, the images were analyzed by NIH ImageJ version 1.52.

### 4.10. Measurement of Cyto Ca^2+^

Fura-2-acetoxymethyl ester (Fura 2-AM, Biotium, Fremont, CA, USA, 50033) was used to detect changes in the Cyto Ca^2+^ as previously described [48]. These experiments were performed in two formats: the cells were treated with 80 μM CoCl_2_ for 4 h; after that, they were either co-cultured with ASCs or cultured in ASC-CM under treatment with 80 μM CoCl_2_ for 24 h. The dynamic changes of Cyto Ca^2+^ were detected by monitoring the fluorescence (Em 535 nm) from Fura 2-AM excited by wavelengths of 340 nm and 380 nm on the multimode microplate reader. The fluorescence intensity of Fura 2-AM was monitored by a Tecan multimode microplate reader every 2 h.

### 4.11. Measurement of ROS Signal

HUVECs expressing roGFP3 were seeded into 24-well-plates (1 × 106 cells/well). One-day post-seeding cells were first treated with either 80 μM CoCl_2_ for 4 h. Afterward, the HUVECs were either co-cultured with ASCs or cultured in ASC-CM for 24 h, and treatment with 80 μM CoCl_2_ was continued for 24 h. The dynamic changes to the ratio of fluorescence intensity emitted by roGFP3 were monitored on the Tecan multimode microplate reader as previously described [47].

### 4.12. Hind Limb Ischemia

In this study, we employed the atherosclerosis-prone apolipoprotein E-deficient (ApoE^−/−^) mice as our animal model. ApoE^−/−^ mice are a well-established model for atherosclerosis. Especially when fed on a Western diet, the mice spontaneously develop atherosclerotic lesions even when on a standard diet, and these lesions progressively exhibit features closely resembling those of human atherosclerosis. Twenty-four 8-week-old male ApoE^−/−^ mice were procured from the Charles River and fed a Western diet (5% cholesterol and 21% fat) for 12 weeks. All experimental procedures on animals were performed according to the EC guideline EC 2010/63/EU and have been approved by the local German government authority (35-9185.81/G[1]239/18). The mice had free access to food and water. Their cages, bedding, water bottles, and aspen shelters were changed weekly. Ambient temperature and relative humidity were kept at 22–23 °C and 50–70%, respectively. Room illumination was automatically set on a 12 h day/night lighting cycle, with lights on at 7:00 a.m. At 20 weeks of age, Hind limb ischemia (HLI) was induced as described in our recent report [49]. Briefly, the mice were anesthetized with a subcutaneous injection (S.C.) of a mixture of midazolam (5 mg/kg, 44856.01.00, Ratiopharm, Ulm, Germany), medetomidine (0.05 mg/mL/kg, 128366-50-7, Cayman Chemical, Ann Arbor, MI, USA), and fentanyl (0.5 mg/kg, 437-38-7, Cayman Chemical, Hamburg, Germany) before all surgical procedures. After 6 min of anesthesia, when mice breathed slowly and stably, hairs on their lower limbs were removed using a depilatory cream (Veet, Heidelberg, Germany). A saline solution was injected subcutaneously to maintain sufficient fluid in the body. Next, the mouse was placed on a 37 °C heating pad and the skin was rubbed with an alcohol scrub. An incision of about 3 to 4 mm was made in the middle of the groin area using pointed-tip forceps and surgical scissors. To increase the surgical field of view, all surgical procedures were performed under a dissecting microscope (Axiovert 100, ZEISS, München, Germany). The superficial femoral artery (FA) was dissected from the femoral vein (FV) and femoral nerve (FN). The FA was ligated proximally and distally and excised. Afterward, the mice were subjected to a double ligation of the femoral artery (DLFA) using Ethicon 7-0 sutures (8776H, PROLENE, Lidingö, Sweden)to simulate an acute-on-chronic model of PAD, as described previously [49]. After the DLFA, the mice were divided into two groups, which included the control and MSCs treatment groups (*n* = 12, for each group) (Figure 2). The animals in the control group received intramuscular 100 μL normal saline injected into 5 different muscle localizations, while the animals in the treatment group received 100 μL fresh from culture ASCs (1 × 10^8^ cells/mL) suspension in the right hind limb (20 μL for each localization) immediately after the DLFA. With regards to postoperative analgesia, Butorphanol (1 mg/kg, S.C., Q8h, Intervet, Unterschleißheim, Germany, 42408-82-2) was given to the mice 24 h after the DLFA. Drinking water was supplemented with metamizole (24 mg/5 mL of water, corresponding to a dose of 200 mg/kg 4 times daily) to maintain the analgesic effect for 2 days following the DLFA procedure. One-day post-operation, magnetic resonance imaging (MRI) scans of the bilateral hind limbs were taken to document the perfusion situation of the proximal and distal femoral artery (FA) (Appendix A).

### 4.13. Plasma and Tissue Collection

The animals were sacrificed 7 days after the DLFA. Mice were euthanized by injection using a combined anesthetic (MMF) half an hour before harvesting. Blood was collected from the inferior vena cava. Plasma was prepared by centrifuging the blood at 1000 rpm for 10 min at 4 °C and stored at −80 °C for biochemical analysis. Both sides of the fresh vastus lateralis (VL) and Gastrocnemius (GM) were rapidly removed after perfusion with 4% PFA and fresh samples were snap-frozen in liquid nitrogen and kept at −80 °C for further analysis.

### 4.14. Histology and IHC Staining

The histological analysis was carried out on the bilateral gastrocnemius muscle (GM) and the aorta with Hematoxylin & Eosin (HE) and CD31 staining via Immunohistochemistry (IHC) as described previously [49]. ImageJ was used to estimate the percentage of microvascular area (CD31 positive area) in five randomly selected fields of view (×40).

### 4.15. Lactate Dehydrogenase (LDH) Assay

The relative levels of LDH in the vastus lateralis muscle (VL) tissue were detected using an LDH kit (ab197000, Abcam, Germany) according to the manufacturer’s guidelines. The 40 mg tissue was homogenized using a T18 digital homogenizer (IKA, Wilmington, NC, USA) at the highest speed of 25,000 rpm for 30 s after which it was kept on ice for 10 min. Subsequently, tissue lysate was centrifuged for 5 min at 4 °C at 10,000× *g*; the supernatant was then collected and added into 96-well-plates (each measuring 50 μL). Hereafter, 50 μL reaction mix supplied by the kit was added to each sample. Then, the fluorescence intensity at 535 nm/587 nm (Ex/Em) was immediately measured by a multimode microplate reader.

### 4.16. Myoglobin Assay

The relative level of Myoglobin (Mb) in the VL tissue was detected using the Mb kit (ab210965, Abcam, Germany) according to the manufacturer’s instructions. Briefly, 50 μL of tissue lysate supernatant obtained from homogenization of VL was added to each well of a 96-well plate and incubated at room temperature for 1 h. After washing, 100 μL of 3,3′,5,5′-tetramethylbenzidine developing solution was added to each well. The plate was incubated for 10 min and the stop solution (50 μL per well) was added. Absorbance was measured at 450 nm on a multi-mode microplate reader.

### 4.17. Statistical Analysis

Statistical analyses were performed using GraphPad Prism 8 (Graphpad Software, San Diego, CA, USA). The results for the different experimental groups were expressed as $\bar{x}$ ± SD. The differences between groups were analyzed by one-way analysis of variance (ANOVA) followed by Tukey’s post hoc correction analysis. Throughout the analysis, a *p*-value < 0.05 was considered statistically significant.

## 5. Conclusions

In this study, we clearly demonstrated that in endothelial cells under prolonged hypoxia, in vivo, the injection of adipose stem cells can alleviate muscle damage caused by ischemia and can enhance the angiogenesis capacity in muscle tissue. In vitro, stem cells enhance the angiogenesis function of endothelial cells, mainly reflected in cell proliferation, migration, differentiation, and structural rearrangement. The homeostasis of calcium ions in endothelial cells interferes with apoptosis under hypoxia and adjusts the balance of ROS generation in endothelial cells. All these provide new ideas and evidence for further exploration of the therapeutic mechanism of adipose stem cells in clinical.

## Figures and Tables

**Figure 1 ijms-24-17197-f001:**
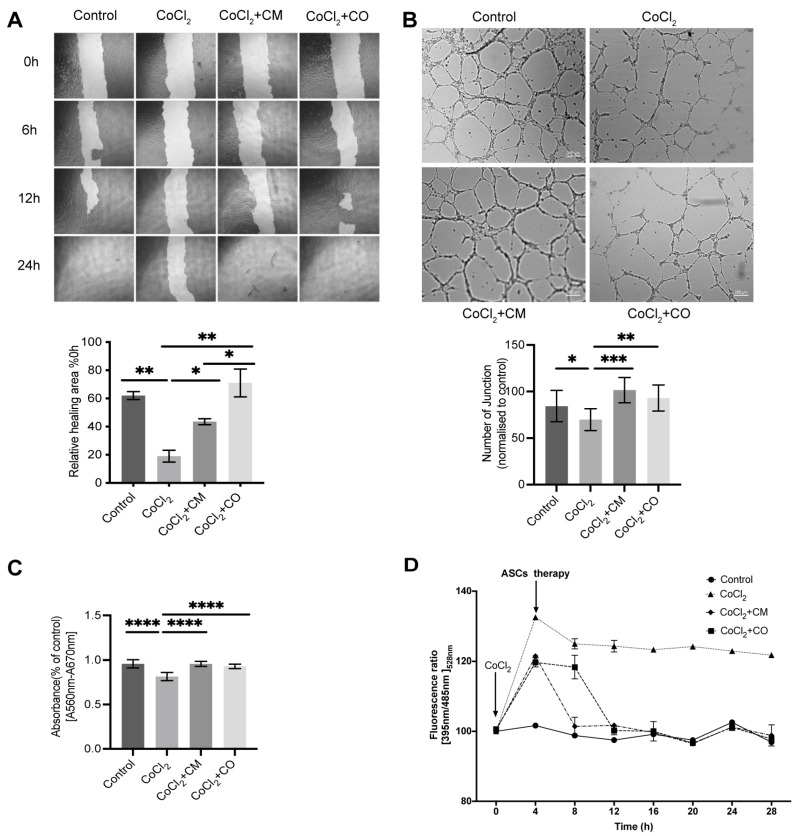
ASCs improved the angiogenic properties of hypoxic HUVECs. HUVECs were treated with 80 µM CoCl_2_ for a period of 4 h; afterward, they were either cultured with ASC-conditioned medium or co-cultured with ASCs. (**A**) The migration of HUVECs at 12 h was analyzed via Scratch assay: CoCl_2_ significantly decreased their migration rate, while the culture of hypoxic HUVECs in ASC-CM and with ASCs (ASC-CO) both restored their migration rate. The gap was quantified via image J, scale bar is 200 μm. (**B**) The number of meshes and junctions significantly reduced in HUVECs treated with CoCl_2_ treatment. In contrast, ASC-CM and ASC-CO significantly restored the HUVECs’ ability to form meshes and junctions despite CoCl_2_ treatment. Cells were normalized, scale bar is 200 μm. (**C**) CoCl_2_ significantly decreased the viability of HUVECs; however, ASC-CM and ASC-CO significantly reversed this effect. (**D**) Treatment with CoCl_2_ for 4 h significantly increased the ROS levels. This effect was reversed by ASC-CM and ASC-CO therapy for 24 h, and the therapeutic effects of ASC-CM were exerted earlier than those of ASC-CO (*n* = 3 replicates; ANOVA, * *p* ≤ 0.05, ** *p* ≤ 0.01, *** *p* ≤ 0.001, **** *p* ≤ 0.0001).

**Figure 2 ijms-24-17197-f002:**
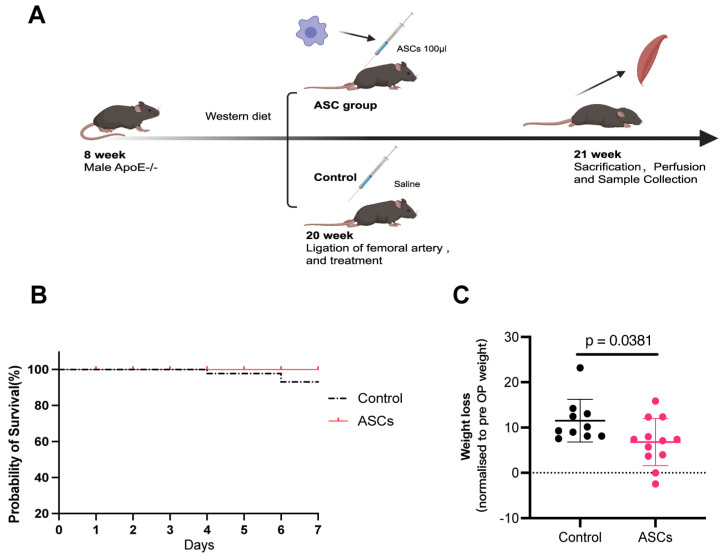
ASCs-treated ApoE^−/−^ mice lost significantly less weight in comparison to the weight lost by control mice one week after critical limb ischemia (CLI). (**A**) The workflow of the animal experiments. The ApoE^−/−^ mice were fed a Western diet containing 4% cholesterol for 12 weeks. CLI was induced in the right leg of the mice via double ligation of the femoral artery (DLFA) at 20 weeks of age, and the segment between the ligatures was excised. The animals in the stem cell group were injected with 100 μL ASCs (1 × 10^8^ cells/mL) suspension in the right hind limb (20 μL for each localization) ASCs, while mice in the control group received 0.9% NaCl solution. (**B**) Comparison of survival rate of mice in the control group and those in the treatment group after the DLFA operation. (**C**) The body weight of each animal was assessed directly prior to the induction of DLFA and seven days thereafter. In the NaCl-treated mice, a significant mean body weight loss of −4.733 ± 2.131 was observed, whereas, in the ASC-treated rats, no significant differences were noted in body weight before and seven days after the induction of DLFA (*n* = 10 for control and *n* = 12 for ASCs group, Unpaired *t*-test; *p* = 0.0381).

**Figure 3 ijms-24-17197-f003:**
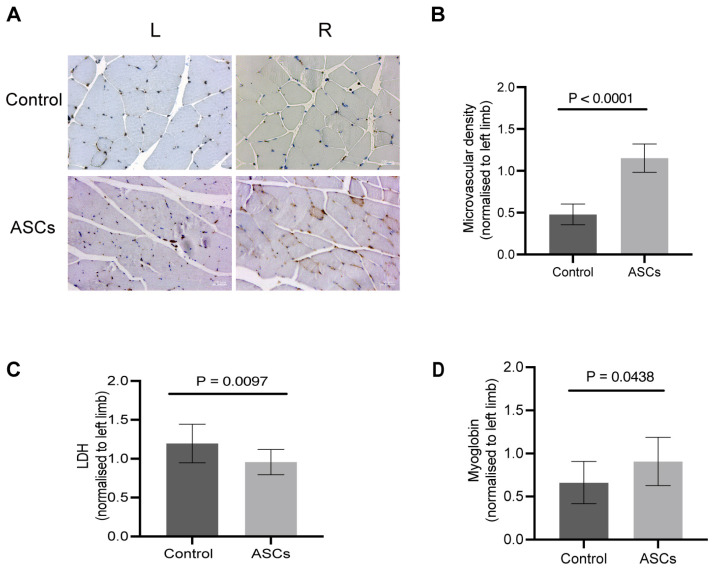
ASCs treatment protects the Gastrocnemius muscle (GM) against DLFA-induced hypoxic injury. (**A**) Representative images showing CD31 staining of bilateral GM of control and ASC-treated mice post DLFA. CD31 positive cells were assessed by immunohistochemistry as described in the materials and methods section (scale bar: 200 μm). (**B**) Quantification of CD31 was performed by quantitative morphometric analysis; the microvascular density (CD31 ratio, right to left leg) in the control group was lower than that in the ASC group. (**C**) Myoglobin and (**D**) LDH were measured, and the ratio in the control group was significantly lower than that in the ASC group; the LDH ratio of GM in the control group was significantly higher than that in the ASC group (presented as ratio of right to left leg) (*n* = 10 for control, ANOVA, *n* = 12 for ASCs).

**Figure 4 ijms-24-17197-f004:**
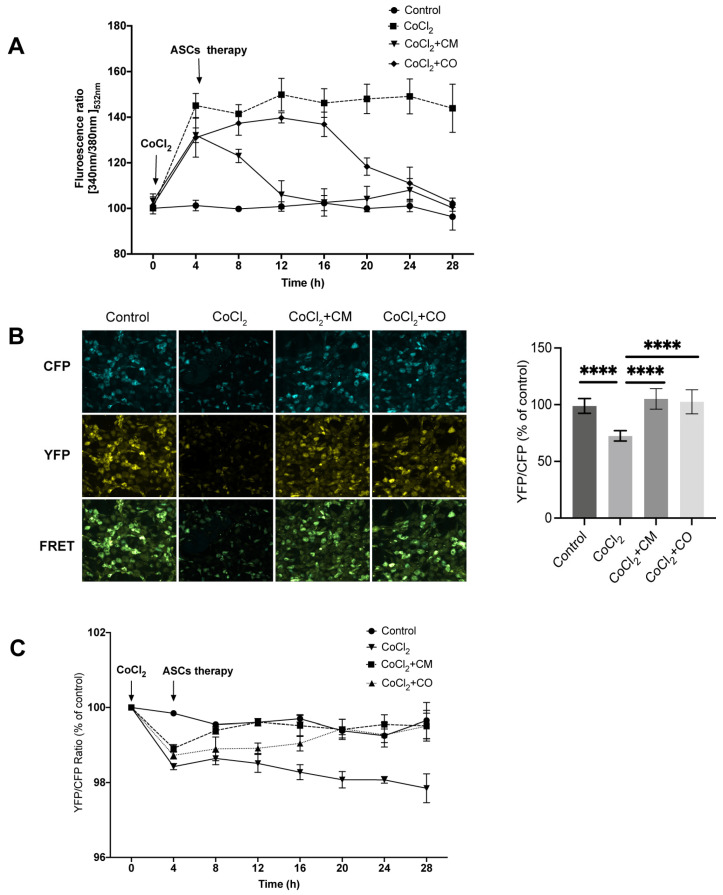
ASC treatment protects the Hypoxia and calcium ion dynamic changes (**A**) Treatment with CoCl_2_ for 4 h significantly increased the Fura-2 ratio. This effect was reversed by ASC-CM and ASC-CO treatment for 24 h. The therapeutic effect of ASC-CM was exerted earlier than that of ASC-CO. (**B**) HUVECs treated with CoCl_2_ displayed a significantly lower FRET ratio than control cells. The ASC-CM and ASC-CO treatments significantly increased the FRET ratio in CoCl_2_ groups and reached a normal level (scale bar is 200 μm). (**C**) Treatment with CoCl_2_ significantly decreased the FRET ratio. This effect was reversed by ASC-CM and ASC-CO treatment in 24 h. The effect of ASC-CM was exerted earlier than that of ASC-CO (*n* = 3 replicates, ANOVA, **** *p* ≤ 0.0001).

**Figure 5 ijms-24-17197-f005:**
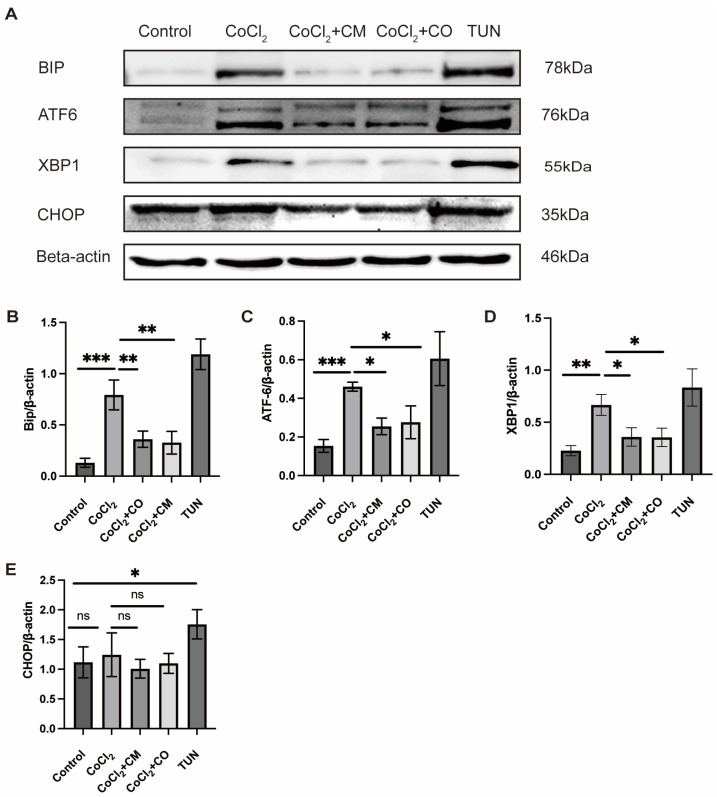
UPR Western blotting in vitro (**A**). (**B**) ASC-CO and ASC-CM reduced the expression of BIP under exposure to CoCl_2_, incubations with tunicamycin, and CoCl_2_ were used as controls (**C**) ASC-CO and ASC-CM reduced the protein expression of ATF6 under CoCl_2_, incubations with tunicamycin, and CoCl_2_ were used as controls (**D**) ASC-CO 57 and ASC-CM reduced the protein expression of XBP1, incubations with tunicamycin, and CoCl_2_. (**E**) The protein expression of CHOP and tunicamycin remained unchanged with ASC-CO and ASC-CM as compared to controls. (*n* = 3, ANOVA ns, not significant; * *p* ≤ 0.05, ** *p* ≤ 0.01, *** *p* ≤ 0.001).

**Figure 6 ijms-24-17197-f006:**
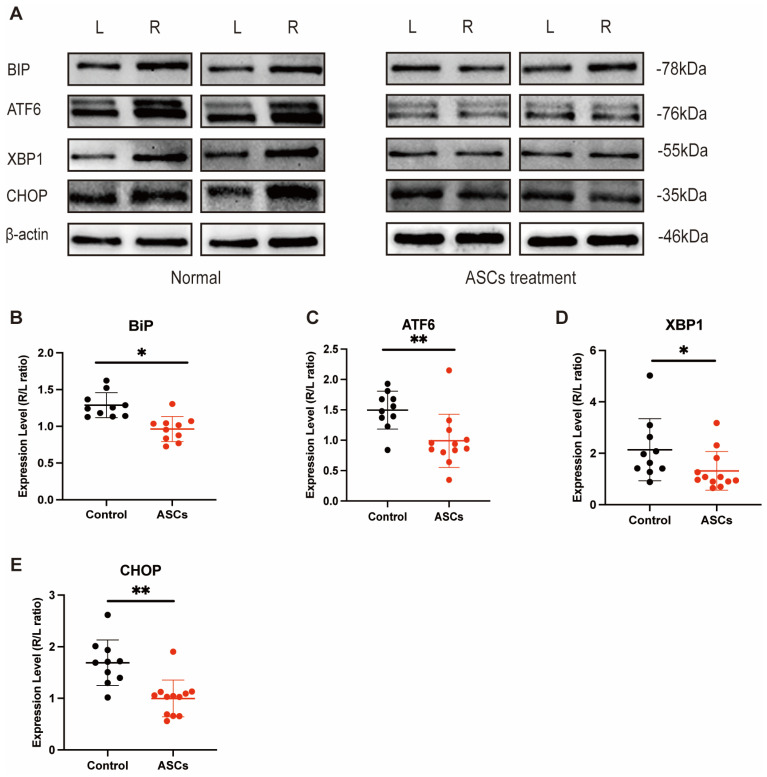
ASCs lead to changes in UPR expression in vivo (**A**) The comparison of the left and right legs—with DLFA- of the two groups of mice are shown (**B**) Ratio between BIP expression between left and right leg with and without ASCs (**C**) Ratio of ATF6 expression between left and right leg with and without ASCs (**D**) Ratio of CHOP expression between left and right leg with and without ASCs (**E**) Ratio of XBP1 expression between left and right leg with and without ASCs (*n* = 3, ANOVA, * *p* ≤ 0.05, ** *p* ≤ 0.01).

## Data Availability

The data that support the findings of this study are available on request from the corresponding author (Vugar Yagublu).

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
