# Peer review of "Adipose-Derived Mesenchymal Stem Cells Protect Endothelial Cells from Hypoxic Injury by Suppressing Terminal UPR In Vivo and In Vitro"

_ijms, 2023, doi:10.3390/ijms242417197_

Round 1

Reviewer 1 Report

Comments and Suggestions for Authors

This manuscript has too many unclear parts.

Moreover, the manuscript is lack of innovation, and experimental design of this manuscript is very poor.

 Thus, the manuscript is not of sufficient quality or novelty to be published in IJMS.

#1:

The definition of ASCs is seriously confused. In Abstract, ASCs is an abbreviation of adipose derived stem cells (line 25). However, in Introduction, ASCs is an abbreviation of adipose stromal cells (line 58).

Adipose derived stem cells and adipose stromal cells are completely different cell type.

#2:

More detailed information of ASCs is needed for understanding the protection of endothelial cells by ASCs from hypoxic injury. ASCs were isolated from human adipose tissues? Does the treatment with ASCs on ischemic muscle tissue in the APOE-/- mice is xenotransplantation model?

#3:

Lines 527-528: “The histological analysis was carried out on the bilateral gastrocnemius muscle (GM) and the aorta….”

There is no description of aorta in neither Results nor Figures. 

#4:

Lines 531-532: “The images were analyzed to collect individual muscle fiber size, central nucleus cell number, and number of adipose cells per image using ImageJ. The mean ratios were plotted and compared.”

There is no description of “individual muscle fiber size, central nucleus cell number, and number of adipose cells per image” in neither Results nor Figures. 

#5:

Lines 523-525: “One day post operation, magnetic resonance imaging (MRI) scans of the bilateral hind limbs were taken to document the perfusion-situation of the proximal and distal femoral artery (FA).”

There is no description of MRI in neither Results nor Figures.

#6:

The reason of application of APOE-/- mice as the hindlimb ischemia model for peripheral artery disease should be explained in Materials and Methods or Results.

Furthermore, APOE-/- mice and ApoE-/- mice (line 227) are mixed in the text.

#7:

There are no detailed explanation of procedures and handling for left and right hindlimbs of hindlimb ischemia model mice in the text.

#8:

In “4. Materials and Methods, 4.14 LDH Assay”

LDH is usually measured in serum as a deviating enzyme in organ damages. The reason of application of homogenized muscle tissues is not clear.

Author Response

Dear Editorial Team,

We sincerely thank you and all reviewers for the feedback on our manuscript entitled “Adipose derived mesenchymal stem cells protect endothelial cells from hypoxic injury by suppressing terminal UPR in vivo and in vitro” (ijms-2674646). We found the comments from editorial members and reviewers are very helpful in improving the manuscript.

In the following, we have addressed the comments/questions in a point-by-point manner where reviewers’ comments are formatted in bold and our replies are formatted in simple text.  

We hope the revised manuscript will find your approval for publication in International Journal of Molecular Sciences.

Thank you very much and looking forward to hearing from you soon!

Sincerely yours,

Dr.Vugar Yagublu

Reviewer 2 Report

Comments and Suggestions for Authors

The manuscript entitled as “Adipose derived mesenchymal stem cells protect endothelial cells from hypoxic injury by suppressing terminal UPR in vivo and in vitro” is a research manuscript that contains in vivo and in vitro models, including co-cultures and transfected cells. The manuscript is well-written and well-structured. Below, the authors can find some suggestions and comments.

Abstract:

Succinct and adequate.

Introduction:

Objective and well-structured, presenting the disease, treatment, ASCs, and hypoxia. As a suggestion, the authors could add more information (summarized) about the assays performed in the manuscript in the last paragraph. Moreover, the authors should include, in a brief format, what was already published in this topic, as well as the innovative character of the present manuscript. As a reader, a disclaim about the innovative part of the research is missing. 

Methods:

Add the number of independent assays in each section.

Disclaimed the ethical aspect of the HUVECs and ACSs isolation, and the in vivo assessment. Good job.

Expression of MSC markers: Add the technique to evaluate it (e.g., flow cytometry) and the most important markers (since these cells were isolated and not purchased), and the passage used.

4.3. Check the superscript and subscript of the text. Additionally, there are several types of available thinCerts, add the type of the membrane and the pore size. In fact, check the entire manuscript for unformatted writing.

Line 421: spell check “transduced”.

4.12 or 4.13: add the timepoints, that is, the period immediately after the DFLA performance and the animal’s sacrifice to carry out the histology, IHC and so on. Add the euthanasia technique in 4.12.

Add the LDH meaning.

As a suggestion, the authors could add a scheme of the in vitro performed assays (like figure 2A) to facilitate the reading and appreciation of the research by the readers. There are 2 cell lines, co-culture, culture medium, and transfected HUVEC cells.

Results:

Line 124: check imageJ spell.

Standardize the terms critical limb ischemia, double ligation of the femoral artery and its abbreviation.

Figure 2 caption: The authors started the caption describing the results of the graph C. As a suggestion, the authors could start with a generic description of all included figures, and then, describe them in the order of appearance.

To facilitate the graph interpretation, authors can standardize the symbols of control and experimental groups of the figures 1D, 4A, 4C, that is, squares representing the same group in all graphs, and so on.

Discussion:

Started with the in vivo results and then, explored the mechanistic using the in vitro results. Moreover, the authors addressed the delay of the co-culture in relation to the conditioned medium. In the authors’ opinion, future research should use conditioned medium injections in instead of ASC?

BiP and BIP – standardize the abbreviation.

Authors competently described the present results using the concepts found in the literature; However, to provide a more comprehensive discussion, authors should compare the obtained results with the literature (if there is any), highlighting the similarities and differences with similar and distinct experimental models.

Conclusion and limitations are adequate.

References seem ok. 

Author Response

(The authors gave the same response as above.)

Reviewer 3 Report

Comments and Suggestions for Authors

The paper entitled “Adipose derived mesenchymal stem cells (ADMCs) protect endothelial cells from hypoxic injury by suppressing terminal UPR in vivo and in vitro” is a study based on ASCs for therapeutic use in peripheral artery disease.

The aim of this study was to evaluate the mechanism of multipotent ADMCs on hypoxic endothelial cells (ECs) and terminal unfolded protein response (UPR) in vitro and in mice hindlimb ischemia model in vivo.

The topic is of clinical interest especially considering the novel therapeutic options of ADMCs that are continuously being researched in all fields of medicine.  

The study has been correctly planned and represents a valid model for future studies in this field. The study provides objective results and is relevant in this field. The conclusions are consistent with the continents presented throughout the text and the main questions have been addressed in an appropriate manner. References are appropriate. The figures and tables are pertinent, and descriptive and assist in describing the results.

Clinical perspectives and implications are lacking in the Discussion and should be included.  The authors should also comment on the potential therapeutic uses in humans and future studies that are needed to gain further insights into the mechanisms and uses of stem cells in treatment.

Comments on the Quality of English Language

Editing can improve the English and flow of the text.

Author Response

(The authors gave the same response as above.)

Round 2

Reviewer 1 Report

Comments and Suggestions for Authors

1)

In “Supplementary Materials”

The description of legend for Figure 1s, “scale bar is 200 µm and 50µm”, may be correct.

However, both scale bars appearing in upper photographs for H&E sections describe 200 µm.

Furthermore, there is no explanation to distinguish between the 4 photographs.

2)

Lines 151: Western diet led to plaque formation in both groups.

Detailed information of “both groups” is needed.

3)

In this paper, important results as in vivo experiments were obtained by administering human ASC cells to mice.

Therefore, the fact of a xenotransplantation must be described clearly in both Abstract and Result sections.

Author Response

Dear Reviewer,

We are grateful for the questions and concerns you have raised in this round of review. Each point has been carefully considered, and we agree that addressing these will further enhance the clarity and depth of our study. We are committed to making the necessary revisions and believe that these changes will provide a more comprehensive understanding of our research.

All the changes which  we have done during the first review prosess are accepted in the Manuscript, so that they are not visible now. All the new changes made in the Manuscrip are now marked accordingly.  

With Best Regards

Vugar Yagublu 
